# Evolutionarily conserved sperm factors, DCST1 and DCST2, are required for gamete fusion

Naokazu Inoue[1]*, Yoshihisa Hagihara[2], Ikuo Wada[1]

[1]Department of Cell Science, Institute of Biomedical Sciences, School of Medicine, Fukushima Medical University, Fukushima, Japan; [2]Biomedical Research Institute, National Institute of Advanced Industrial Science and Technology (AIST), Ikeda, Japan

**Abstract** To trigger gamete fusion, spermatozoa need to activate the molecular machinery in which sperm IZUMO1 and oocyte JUNO (IZUMO1R) interaction plays a critical role in mammals. Although a set of factors involved in this process has recently been identified, no common factor that can function in both vertebrates and invertebrates has yet been reported. Here, we first demonstrate that the evolutionarily conserved factors dendrocyte expressed seven transmembrane protein domain-containing 1 (DCST1) and dendrocyte expressed seven transmembrane protein domain-containing 2 (DCST2) are essential for sperm–egg fusion in mice, as proven by gene disruption and complementation experiments. We also found that the protein stability of another gamete fusion-related sperm factor, SPACA6, is differently regulated by DCST1/2 and IZUMO1. Thus, we suggest that spermatozoa ensure proper fertilization in mammals by integrating various molecular pathways, including an evolutionarily conserved system that has developed as a result of nearly one billion years of evolution.

*For correspondence:
n-inoue@fmu.ac.jp

**Competing interests:** The authors declare that no competing interests exist.

## Introduction

Gamete recognition and fusion in mammals are considered to occur through a complex intermolecular interaction in which izumo sperm–egg fusion 1 (IZUMO1), sperm acrosome associated 6 (SPACA6), transmembrane protein 95 (TMEM95), fertilization influencing membrane protein (FIMP) and sperm–oocyte fusion required 1 (SOF1) on the sperm side and JUNO (also known as IZUMO1 receptor) and cluster of differentiation 9 (CD9) on the ovum side are all involved in membrane fusion, as proven by gene disruption (*Barbaux et al., 2020*; *Bianchi et al., 2014*; *Fujihara et al., 2020*; *Inoue et al., 2005*; *Kaji et al., 2000*; *Lamas-Toranzo et al., 2020*; *Le Naour et al., 2000*; *Lorenzetti et al., 2014*; *Miyado et al., 2000*; *Noda et al., 2020*). Particularly, the IZUMO1–JUNO interaction is likely to be essential for triggering gamete fusion (*Aydin et al., 2016*; *Inoue et al., 2015*; *Inoue et al., 2013*; *Ohto et al., 2016*); however, it is unknown how these factors contribute to this process (*Bianchi and Wright, 2020*).

Here, we show that an osteoclast fusion-related factor, dendrocyte expressed seven transmembrane protein (DC-STAMP) (*Kukita et al., 2004*; *Yagi et al., 2005*), homologue DC-STAMP domain-containing 1 (DCST1), and its paralogue DCST2, which show testis and haploid-specific expression (*Figure 1D*), are indispensable for fertilization in mice. Mouse DCST1 is considered to be an orthologue of *Caenorhabditis elegans* spermatogenesis-defective 49 (SPE-49) (*Wilson et al., 2018*) and *Drosophila* SNEAKY (*Wilson et al., 2006*), whereas mouse DCST2 is considered to be an orthologue of *C. elegans* SPE-42 (*Kroft et al., 2005*) and *Drosophila* DCST2, although the numbers of putative transmembrane regions appear different (*Figure 1A,B*). Previously, these factors have been shown to be essential for fertilization. In fact, phylogeny analysis revealed that DCST1 and DCST2 belong

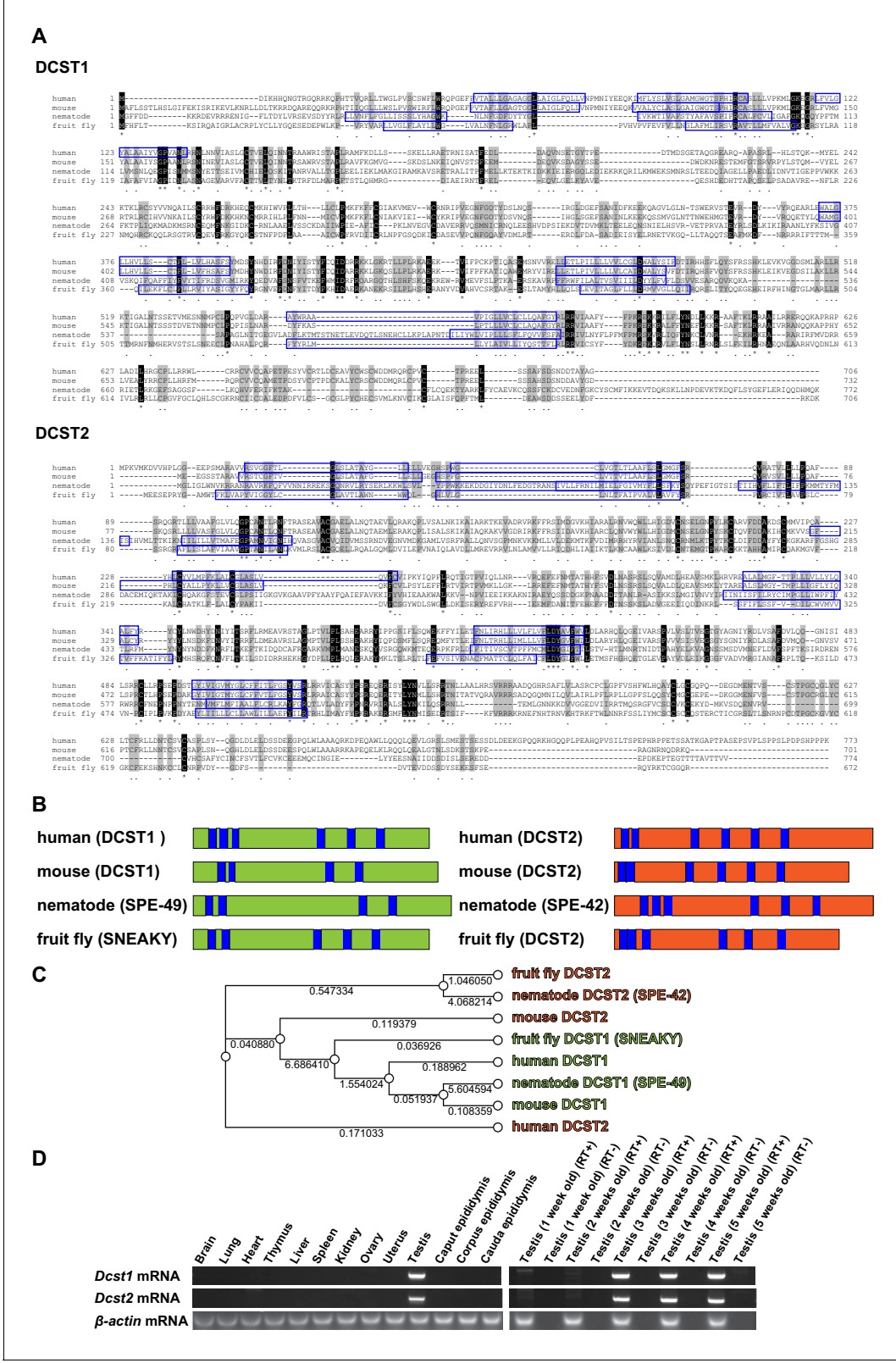

**Figure 1.** Alignments of dendrocyte expressed seven transmembrane protein domain-containing 1 (DCST1) and dendrocyte expressed seven transmembrane protein domain-containing 2 (DCST2) amino acid sequences, and mRNA expression profiles. (**A**) Sequence alignments of DCST1 in *Homo sapiens* (human; GenBank: NP_689707.2), *Mus musculus* (mouse; GenBank: NP_084250.1), *Caenorhabditis elegans* (nematode; GenBank: NP_001343858.1), and *Drosophila guanche* (fruit fly; GenBank: XP_034122086.1) are shown. Sequence alignments of DCST2 in *Homo sapiens* (human;

*Figure 1 continued on next page*

*Figure 1 continued*

GenBank: NP_653223.2), *Mus musculus* (mouse; GenBank: NP_001357782.1), *Caenorhabditis elegans* (nematode; GenBank: NP_872213.4), and *Drosophila mauritiana* (fruit fly; GenBank: XP_033160419.1) are shown. Identical and similar amino acid residues are represented as asterisks and dots, respectively. The residues of putative transmembrane domains are indicated by blue boxes. (B) Schematic diagram of DCST1 and DCST2 in various species. The blue area indicates the putative transmembrane domain. (C) Phylogeny analysis of DCST1 and DCST2. Maximum likelihood phylogenetic tree was constructed using RAxML with 1000 bootstraps through Genetyx software. The distance is shown as a numerical value. (D) Tissue and age-dependent expression profile of *Dcst1* and *Dcst2* mRNA in mice. *β-actin* mRNA was used as an internal control.

to different clades (*Figure 1C*). This supports the finding that DCST2 appears to be closer to ancestor molecules than DCST1, because DCST1 is considered to have arisen after branching from DCST2 (*Figure 1C*).

For the first time, we here provide evidence that a common set of related factors between vertebrates and invertebrates actively participates in fertilization and appears to promote gamete merging in mammals.

## Results and discussion

Since *Dcst1* and *Dcst2* are transcribed in different directions so as to face each other (*Figure 2A*), we first produced *Dcst1/2* doubly-disrupted mouse lines using the homologous recombination system in embryonic stem cells, replacing *Dcst1* exon 1–3 and *Dcst2* exon 1–4, including the promoter region with the neomycin-resistance gene (*Neo$^r$*) (*Figure 2A,B*). The *Dcst1/2$^{-/-}$* male mice, but not female, became completely infertile (*Figure 3E*). When eggs were collected from the oviducts of the female mice 8 hr after coitus with a *Dcst1/2$^{-/-}$* male, many zona-penetrated spermatozoa that had not been fertilized, and possessed a normal IZUMO1 localization upon acrosome reaction, were seen in the perivitelline space of oocytes (*Figure 3A*). There was no difference in sperm motility between the *Dcst1/2$^{-/-}$* and wild-type mice (motility was measured 120 min after incubation by computer-assisted sperm motility analysis in CEROS; mean ± s.e.m = 97.0 ± 0.73% in wild-type [1522 spermatozoa, six individuals] and 96.8 ± 0.87% in *Dcst1/2$^{-/-}$* [1425 spermatozoa, six individuals]). Therefore, it is reasonable to assume that there are no disturbances in sperm migration into the oviduct, acrosome reaction, and zona penetration (*Figure 3A*). More precisely, gamete fusion assay showed that *Dcst1/2$^{-/-}$* spermatozoa were capable of binding to the plasma membranes of oocytes, but no *Dcst1/2$^{-/-}$* spermatozoa had fused with the oocytes (*Figure 3B*). Since the syngamy was restored by bypassing with intracytoplasmic sperm injection (ICSI) using *Dcst1/2$^{-/-}$* spermatozoa (numbers of pups born from *Dcst1/2$^{+/-}$* and *Dcst1/2$^{-/-}$* spermatozoa per survived transplanted 2 cell embryos were 8/27 [30%] and 17/48 [35%], respectively), DCST1/2 should be an essential part of gamete fusion machinery. Furthermore, the oocyte fusion-related factors JUNO and CD9 were found to be concentrated in regions, at which acrosome-reacted (AR) spermatozoa from *Dcst1/2*-deficient mice labeled with α-IZUMO1 antibody were bound, same as wild-type spermatozoa (*Figure 3C*, *Inoue et al., 2020*), suggesting that *Dcst1/2$^{-/-}$* spermatozoa are capable of recruiting the oocyte factors at the contact site; however, these spermatozoa fail to proceed to membrane fusion with oocytes.

In order to know which factor plays a more crucial role in fertilization, we next produced solo *Dcst1$^{-/-}$* and *Dcst2$^{-/-}$* mouse lines using CRISPR/Cas9-mediated gene disruption, and assessed male fertilization ability simultaneously with performed transgenic rescued experiments (*Figure 2A,B*). In *in vitro* fertilization (IVF), as expected, all of the *Dcst1/2$^{-/-}$*, *Dcst1$^{-/-}$* and *Dcst2$^{-/-}$* spermatozoa failed to fertilize the eggs, where the average fertilization rates of *Dcst1/2$^{-/-}$*, *Dcst12$^{-/-}$*, *Dcst2$^{-/-}$* spermatozoa were 0%, 3.3%, and 0.8%, respectively (*Figure 3D*). The impaired fertilization step undoubtedly followed zona penetration, because motile *Dcst1/2*-deficient spermatozoa penetrated the zona pellucida and accumulated in the perivitelline space of the oocytes (*Video 1*). It is noteworthy that the spermatozoa from the single gene transgenic rescued mice (*Dcst1*-TG or *Dcst2*-TG) with *Dcst1/2* knockout genetic backgrounds were unsuccessful in fertilization; however, the spermatozoa from the double transgenic mice (*Dcst1/2$^{-/-}$Dcst1/2*-TG) had normal fertility compared to that of the *Dcst1/2$^{+/-}$* mice (*Figure 3D*). Regarding the numbers of offspring, which is the final outcome of fertility, all *Dcst1/2$^{-/-}$*, *Dcst1$^{-/-}$* and *Dcst2$^{-/-}$* males were completely sterile despite normal mating behavior with ejaculation and vaginal plug formation, whereas the double transgenic rescued males (*Dcst1/*

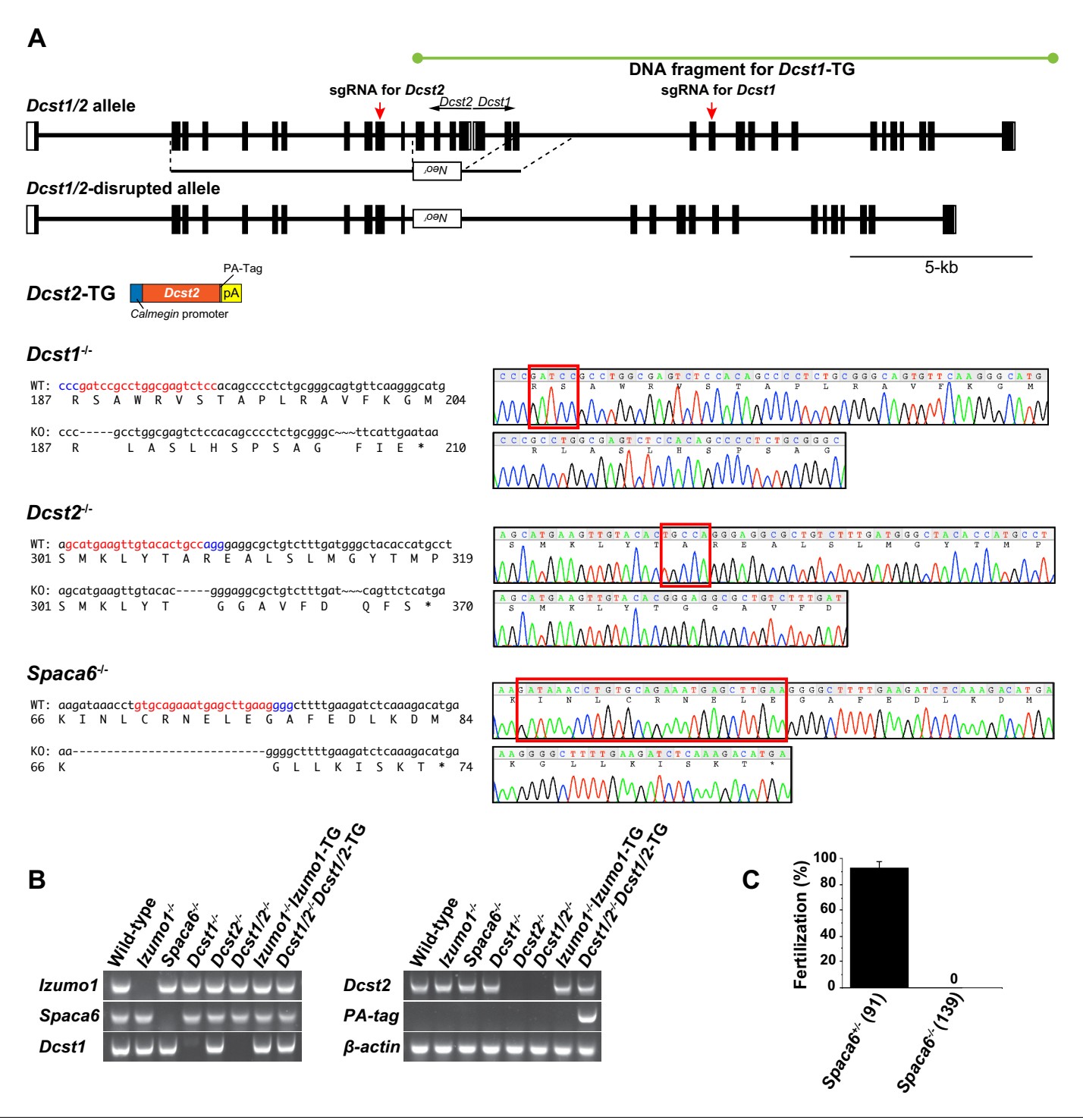

**Figure 2.** Strategy for *Dcst1* and *Dcst2* genes disruption and its validation. (**A**) Targeted disruption of the *Dcst1/2*, *Dcst1*, *Dcst2*, and *Spaca6* genes, and constructs for *Dcst1* and *Dcst2* transgenic mice. Complete structures of the wild-type mouse *Dcst1/2* alleles are shown. Exons and introns are represented by boxes and horizontal lines, respectively. An open reading frame for *Dcst1 and Dcst2* genes is shown in black. For the targeted disruption of mouse *Dcst1/2*, the neomycin-resistance gene (*Neoʳ*) was inserted between *Dcst1* exon 3 and *Dcst2* exon 4. A herpes simplex virus thymidine kinase gene was introduced into the targeting construct for negative selection. For the gene complementation experiments, transgenic mouse lines expressing dendrocyte expressed seven transmembrane protein domain-containing 1 (DCST1), in which full-length genome DNA fragment including all promoter region, exons, and introns was incorporated, and dendrocyte expressed seven transmembrane protein domain-containing 2 (DCST2), in which full-length cDNA with a PA-tag sequence was inserted between a testis-specific (pachytene spermatocyte to spermatid stage)

*Figure 2 continued on next page*

*Figure 2 continued*

*Calmegin* promoter and rabbit *β-globin* polyadenylation signal, were produced. For genome-editing of *Dcst1*, *Dcst2*, and *Spaca6* using the CRISPR/Cas9 system, the red and blue letters show the target and protospacer adjacent motif (PAM) sequence, respectively (left panels). A DNA sequence chromatogram shows that $Dcst1^{-/-}$, $Dcst2^{-/-}$ and $Spaca6^{-/-}$ had 5-base, 5-base, and 28-base deletions, respectively. The deleted sequences are boxed in red (right panels). (B) Validation of gene disruption by reverse transcription polymerase chain reaction (RT-PCR). All primer sets employed wild-type mRNA-specific oligonucleotide sequences except for PA-tag. As a result, all gene disruptions and transgenes were confirmed appropriately. *β-actin* mRNA was used as an internal control. (C) *In vitro* fertilization analysis using $Spaca6^{+/-}$ and $Spaca6^{-/-}$ spermatozoa (n = 4 and 4, respectively). The fertilization rate was evaluated at the two-cell stage embryo. The numbers in parentheses indicate the numbers of oocytes used. The error bars represent standard error of the mean (s.e.m.).

The online version of this article includes the following source data for figure 2:

**Source data 1.** IVF in SPACA6 heterozygous and homozygous KO spermatozoa.

$2^{-/-}Dcst1/2$-TG) showed normal litter sizes, consistent with IVF (*Figure 3D,E*). From these results, we conclude that both DCST1 and DCST2 are required for establishing the gamete fusion leading to embryogenesis.

Finally, we investigated a protein profile of sperm factors such as IZUMO1 and SPACA6, which are essential factors for gamete recognition and fusion (*Figure 2A,C*; *Barbaux et al., 2020*; *Inoue et al., 2005*; *Lorenzetti et al., 2014*; *Noda et al., 2020*). As a result, in *Izumo1*, *Spaca6*, *Dcst1/2*, *Dcst1*, and *Dcst2*-deficient spermatozoa, SPACA6 typically disappears from the mature spermatozoa, whereas the IZUMO1 protein remains, except in *Izumo1*-null spermatozoa (*Figure 3F*). However, loss of SPACA6 was recovered by *Izumo1* complementation ($Izumo1^{-/-}Izumo1$-TG) (*Figure 3F*). Thus, IZUMO1 and SPACA6 are likely to be cooperative factors. Unexpectedly, although *Dcst1/2*-deficient male fertility could be restored in the *Dcst1/2* transgenic rescued mice (*Figure 3D,E*), retrieval of SPACA6 was not observed (*Figure 3F*). While we do not have the data to explain this, the result implies that SPACA6 seems to be dispensable for fertilization in $Dcst1/2^{-/-}Dcst1/2$-TG spermatozoa. In this context, it is interesting that DCST1 has a ubiquitin ligase activity (*Nair et al., 2016*). It is conceivable that precise temporal expression of putative substrates negatively regulated by ubiquitin-mediated degradation by DCST1/2 during spermatogenesis has to be required for SPACA6 protein stability, which could have been impaired in the *Dcst1/2* complementation.

It should be emphasized that DCST1/2 are the first identified well-conserved factors from human to nematode or fruit fly that are actively involved in gamete recognition and fusion. Although the detailed molecular mechanisms underlining gamete recognition and fusion are still unknown, new insight into molecular behavior and interaction through DCST1 and DCST2 involvement would greatly advance our understanding of elaborate common molecular mechanisms in mysterious fertilization in a sexually reproducing organism.

# Materials and methods

### Key resources table

| Reagent type (species) or resource | Designation | Source or reference | Identifiers | Additional information |
|---|---|---|---|---|
| Strain, strain background (*Mus musculus*) | *Dcst1/2* KO | This article | | Deposited in RIKEN BRC (ID RBRC05733) |
| Strain, strain background (*Mus musculus*) | *Dcst1* KO | This article | | Deposited in RIKEN BRC (ID RBRC10275) |
| Strain, strain background (*Mus musculus*) | *Dcst2* KO | This article | | Deposited in RIKEN BRC (ID RBRC10366) |
| Strain, strain background (*Mus musculus*) | *Spaca6* KO | This article | | Deposited in RIKEN BRC (ID RBRC10367) |

*Continued on next page*

*Continued*

| Reagent type (species) or resource | Designation | Source or reference | Identifiers | Additional information |
|---|---|---|---|---|
| Strain, strain background (*Mus musculus*) | *Izumo1* KO | *Inoue et al., 2005* | RRID MGI_3576518 | |
| Strain, strain background (*Mus musculus*) | *Dcst1* TG | This article | | Deposited in RIKEN BRC (ID RBRC05856) |
| Strain, strain background (*Mus musculus*) | *Dcst2* TG | This article | | |
| Strain, strain background (*Mus musculus*) | *Izumo1* TG | *Inoue et al., 2005* | | |
| Antibody | Anti-mouse SPACA6 (rabbit polyclonal) | This article | | (WB: 1:500) |
| Antibody | Anti-mouse IZUMO1 (rat monoclonal) | *Inoue et al., 2013* | Mab125 | Gift from Dr. Ikawa (IF: 0.5 μg ml$^{-1}$) |
| Antibody | Anti-mouse IZUMO1 (rat monoclonal) | *Inoue et al., 2015* | Mab18 | (WB: 1 μg ml$^{-1}$) |
| Antibody | Anti-mouse FR4 (JUNO) (rat monoclonal) | BioLegend | TH6, RRID AB_1027724 | (IF: 0.25 μg ml$^{-1}$) |
| Antibody | Anti-mouse CD9 (rat monoclonal) | BioLegend | MZ3, RRID AB_1279321 | (IF: 0.5 μg ml$^{-1}$) |
| Antibody | Anti-mouse EMMPRIN (BASIGIN) (goat polyclonal) | Santa Cruz Biotechnology | G-19, RRID AB_2066959 | (WB: 0.2 μg ml$^{-1}$) |
| Recombinant DNA reagent | Plasmid pX330 vector | Addgene | RRID Addgene_42230 | |
| Sequence-based reagent | Primers | This article | | Detailed in 'Materials and methods' |
| Peptide, recombinant protein | SPACA6$_{46-139}$ | This article | | Detailed in 'Materials and methods' |

### Mice

C57BL/6, B6D2F1, and ICR mice (*Mus musculus*) were purchased from Japan SLC Inc, and IZUMO1 knockout and transgenic mice (*Inoue et al., 2005*) were kindly provided by Osaka University. All animal studies were approved by the Animal Care and Use Committee of Fukushima Medical University, Japan, and performed under the guidelines and regulations of Fukushima Medical University.

### Antibodies

An α-mouse IZUMO1 monoclonal antibody (Mab125) was conjugated to Alexa Fluor 546. FITC–conjugated α-mouse CD9 monoclonal (MZ3) and Alexa Fluor 647–labeled α-mouse JUNO monoclonal (TH6) antibodies were purchased from BioLegend. Unlabeled α-mouse BASIGIN polyclonal antibody (sc-9757) was purchased from Santa Cruz Biotechnology.

### Generation of *Dcst1/2* double knockout mice

A targeting vector was constructed using modified pNT1.1 containing neomycin-resistance gene as a positive selection marker and herpes simplex virus thymidine kinase gene as a negative selection marker (http://www.ncbi.nlm.nih.gov/nuccore/JN935771). A 1.7 kb *Not* I-*Xho* I fragment and a 6.8 kb *Pac* I-*Mfe* I fragment were inserted as short and long arms, respectively. Embryonic stem cells derived from 129/Sv were electroporated with *Cla* I digested linearized DNA. Of the 192 G418-resistant clones, five had undergone homologous recombination correctly. Two targeted cell lines were

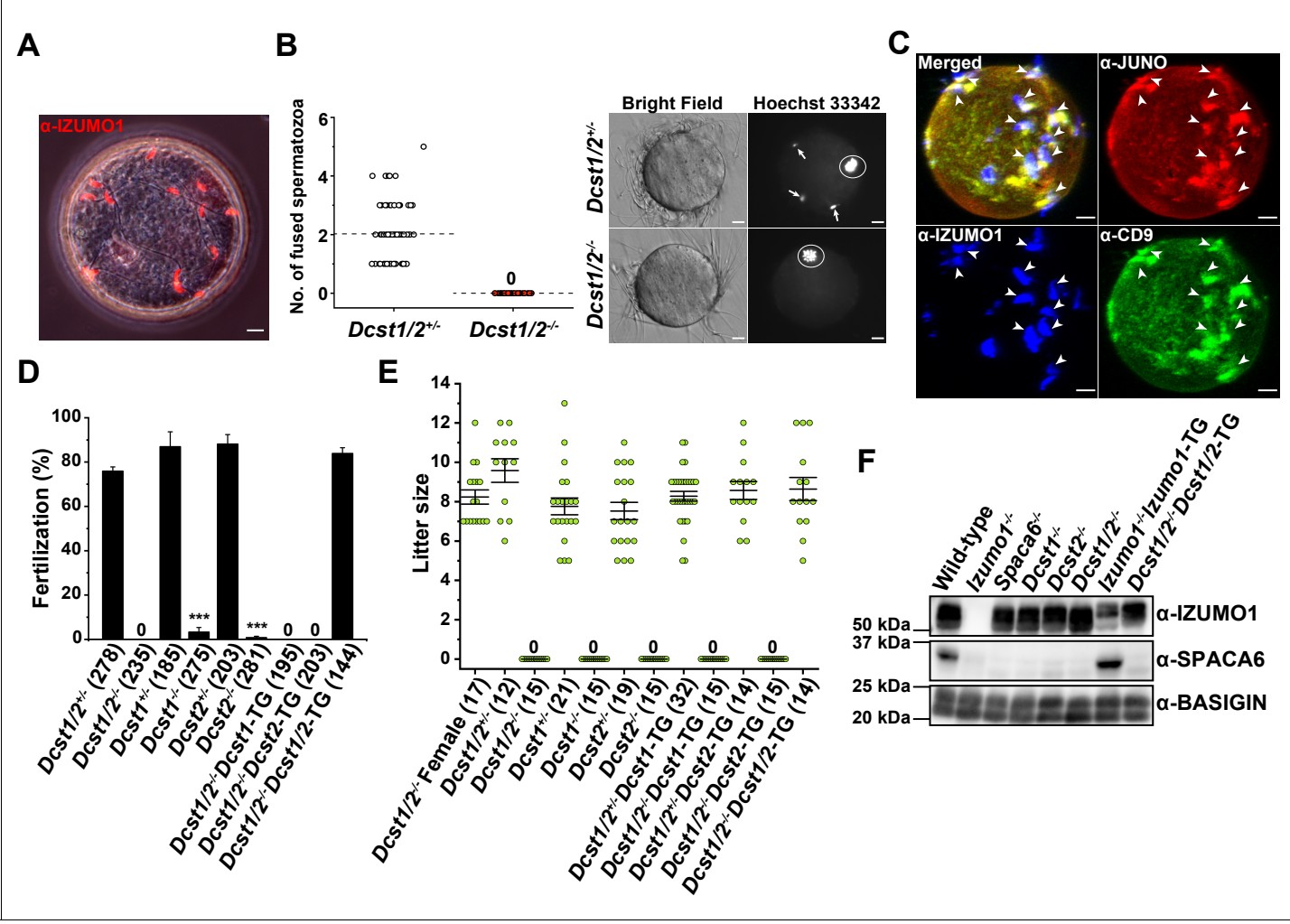

**Figure 3.** Identification of new gamete fusion-related factors dendrocyte expressed seven transmembrane protein domain-containing 1 (DCST1) and dendrocyte expressed seven transmembrane protein domain-containing 2 (DCST2). (**A**) Accumulation of many acrosome-reacted (AR) IZUMO1 positive spermatozoa (red) in the perivitelline space of eggs recovered from the oviducts of females 8 hr after coitus with a *Dcst1/2*[-/-] male. Scale bar 10 μm. (**B**) Gamete fusion assay of *Dcst1/2*-disrupted spermatozoa. Gamete fusion was visualized with Hoechst 33342 dye transfer. The numbers of fused spermatozoa per oocyte from four independent experiments are shown. The total numbers of oocytes examined in *Dcst1/2*[+/-] and *Dcst1/2*[-/-] were 77 and 80, respectively (left graph). The broken lines indicate the average. The arrows and the circles in the representative photo show fused spermatozoa and meiosis metaphase II chromosomes, respectively (right picture). Scale bar 10 μm. (**C**) Localization of oocyte CD9 and JUNO upon *Dcst1/2*[-/-] sperm attachment. Sperm–oocyte interaction was visualized by staining with 0.25 μg ml[−1] TH6–Alexa647 (JUNO: red), 0.5 μg ml[−1] MZ3–FITC (CD9: green) and 0.5 μg ml[−1] Mab125–Alexa546 (IZUMO1: blue). The eggs were fixed 30 min after insemination of *Dcst1/2*[-/-] spermatozoa. The arrowheads indicate the sites where JUNO and cluster of differentiation 9 (CD9) were concentrated in response to the AR spermatozoa (IZUMO1 positive: blue) bound to the oolemma. Scale bar 10 μm. (**D**) *In vitro* fertilization assay using *Dcst1/2*[+/-], *Dcst1/2*[-/-], *Dcst1*[+/-], *Dcst1*[-/-], *Dcst2*[+/-], *Dcst2*[-/-], *Dcst1/2*[-/-]*Dcst1*-TG, *Dcst1/2*[-/-]*Dcst2*-TG, and *Dcst1/2*[-/-]*Dcst1/2*-TG spermatozoa (n = 5, 5, 6, 7, 8, 8, 6, 6, and 6, respectively). The fertilization rate was evaluated at the two-cell stage embryo. The numbers in parentheses indicate the numbers of oocytes used. The error bars represent s.e.m. ***p<0.001 (paired two-tailed Student's t-test, compared with each heterozygous mutants). (**E**) Fecundity of DCST1/2-related deficient male and female mice. The numbers in parentheses indicate the numbers of mating pairs. Regarding *Dcst1/2*[-/-], *Dcst1*[-/-], *Dcst2*[-/-], *Dcst1/2*[-/-]*Dcst1*-TG, and *Dcst1/2*[-/-]*Dcst2*-TG males, the numbers of vaginal plug formations were counted. The error bars represent s.e.m. (**F**) Western blot analysis of various types of mouse lines with 1 μg ml[−1] Mab18 (IZUMO1) and 1:500 SPACA6 anti-serum. Each lane was equally applied with 20 μg sperm lysates. BASIGIN was used as a loading control. The online version of this article includes the following source data for figure 3:

**Source data 1.** Gamete fusion assay, IVF and litter size in DCST1/2-related mutants.

injected into C57BL/6 mice derived blastocysts, resulting in the birth of male chimeric mice. These mice were then crossed with C57BL/6 to obtain heterozygous mutants. The mice used in the present study were the offspring of crosses between F1 and/or F2 generations. The PCR primers used for genotyping were as follows: 5'-ATTCTTCCTCTCCCTTTCGGACATC-3' and 5'-GCTTGCCGAATATCATGGTGGAAAATGGCC-3' for the short arm side of the mutant allele; 5'-ATTCTTCCTCTCCCTTTCGGACATC-3' and 5'-CCAGGGGATCCAACACCCTT-3' for the short arm side of the wild-type allele; 5'-TCTGTTGTGCCCAGTCATAGCCGAATAGCC-3' and 5'-CTGGAAGTGTCTTCCGAGTGCCACTCCACA-3' for the long arm side of the mutant allele; and 5'-CCCTAGGCTGTGAAGTGTTGATGAG-3' and 5'-CTGGAAGTGTCTTCCGAGTGCCACTCCACA-3' for the long arm side of the wild-type allele. The *Dcst1/2*-disrupted mouse line was submitted to RIKEN BioResource Research Center (https://web.brc.riken.jp/en/), and is available to the scientific community (accession number: RBRC05733).

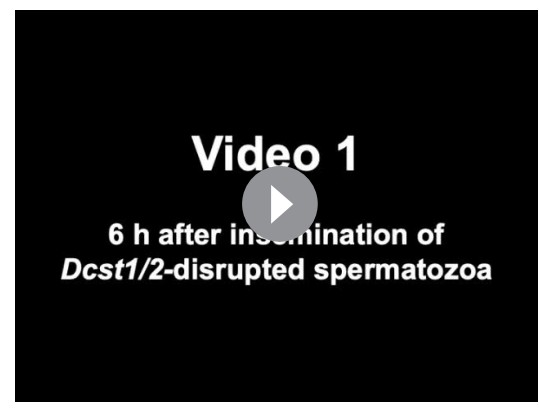

**Video 1.** 6 hr after insemination of *Dcst1/2*-disrupted spermatozoa. The first and second half of the video show *Dcst1/2* heterozygous and homozygous KO spermatozoa insemination, respectively. https://elifesciences.org/articles/66313#video1

## Generation of *Dcst1⁻ᐟ⁻*, *Dcst2⁻ᐟ⁻*, and *Spaca6⁻ᐟ⁻* mice

Pairs of oligonucleotides (5'-CACCGGGAGACTCGCCAGGCGGATC-3' and 5'-AAACGATCCGCCTGGCGAGTCTCCC-3' for DCST1, 5'-CACCGGCATGAAGTTGTACACTGCC-3' and 5'-AAACGGCAGTGTACAACTTCATGCC-3' for DCST2, and 5'-CACCGGTGCAGAAATGAGCTTGAAG-3' and 5'-AAACCTTCAAGCTCATTTCTGCACC-3' for SPACA6) as guide sequences were annealed and cloned into a *Bbs* I cut pX330 vector (Addgene plasmid #42230). pX330 plasmid DNA was purified and diluted into 5 mM Tris-HCl pH 7.4/0.1 mM EDTA of a final 5 ng μl⁻¹ concentration (*Mashiko et al., 2013*). Plasmid DNA was injected into the pronuclei of fertilized eggs with glass capillaries attached to a micromanipulator (FemtoJet, Eppendorf). After microinjection, the zygotes were transferred and cultured in KSOM (ark-resource) at 37°C with 5% $CO_2$, and two-cell embryos were transferred into pseudopregnant ICR female mice. F3 and F4 generation mice were used for the experiments, and all genotyping was performed by PCR on tail-tip DNA using: DCST1, 5'-CATCAGCCAGTAGCCCCAAA-3' and 5'-AGATCAACAACACCCGATCC-3' for wild-type or 5'-CAGATCAACAACACCCGCCT-3' for KO as forward and reverse primers, respectively, and DCST2, 5'-TAGAGAAAGGCAGAAAGTCCAGAC-3' and 5'-AAGACAGCGCCTCCCTGGCA-3' for wild-type or 5'-AAAGACAGCGCCTCCCGTGT-3' for KO and as forward and reverse primers, respectively. SPACA6, 5'-TTTTCTGTCTCTGTCTCTTATGCCAA-3' and 5'-TTCAAAAGCCCCTTCAAGCTCA-3' for wild-type or 5'-AGATCTTCAAAAGCCCCTTGGTTT-3' were used for KO and as forward and reverse primers, respectively. DCST1, DCST2, and SPACA6 knockout mice were deposited at the RIKEN BioResource Research Center (https://web.brc.riken.jp/en/) (accession numbers RBRC10275, RBRC10366 and RBRC10367, respectively).

## Production of DCST1 and DCST2 transgenic mice

*Dcst1* gene fragments were amplified by PCR using a bacterial artificial chromosome (BAC) DNA clone (RP24-185C5) as template DNA. The DNA fragments were ligated into StrataClone PCR Vector pSC-B-amp/kan (Agilent Technologies). Until injection, a 17.5 kb DNA fragment that included whole *Dcst1* gene was digested with *Not* I and *Sal* I (*Figure 2A*). For DCST2, we designed PA-tagged *Dsct2*, to be inserted between the *Calmegin* promoter (testis specific promoter) and a rabbit *β-globin* polyadenylation signal (*Figure 2A*). The transgenic mouse lines were produced by injecting *Dcst1* gene or *Dsct2-PA* DNA fragments into the pronuclei of fertilized eggs. For the detection of transgene, PCR was performed on tail-tip DNA using following primer sets: 5'-GGCCGCTCTAGAACTAGTGGAT-3' and 5'-GGTGTTTGTGTTACTGAAGTCACTG-3' for DCST1-TG specific amplification,

and 5′-CCTTCCTGCGGCTTGTTCTCT-3′ and 5′-GGCTATGTGTTTCACGGCAT-3′ for DCST2-TG specific amplification.

## RT-PCR

Total RNA was extracted from a mouse testis using NucleoSpin RNA Plus and treated with rDNase (MACHEREY-NAGEL). First-strand cDNA synthesis was performed using a PrimeScriptII 1 st strand cDNA Synthesis Kit with Oligo $(dT)_{15}$ primer (Takara Bio). The following primer pairs were designed to amplify each mRNA: IZUMO1 wild-type specific primer pair, forward (5′-CCTATTTGGGGGCCG TGGAC-3′) and reverse (5′-GTCCAGGACCAGGTCTTCCGAGCGA-3′); SPACA6 wild-type specific primer pair, forward (5′-ACCCGGAGACCCTCTGTTAT-3′) and reverse (5′-TTCAAAAGCCCC TTCAAGCTCA-3′); DCST1 wild-type specific primer pair, forward (5′-CTCATCAACACTTCACAGCC TAGGG-3′) and reverse (5′-GGAGACTCGCCAGGCGGATC-3′); DCST2 wild-type specific primer pair, forward (5′-ATGCCGTGAAACACATAGCC-3′) and reverse (5′-AAGACAGCGCCTCCCTGGCA-3′); PA-tag specific primer pair, forward (5′-ATGGAGAACTTTGTGTCCTGCAGTA-3′) and reverse (5′-ACCACATCATCTTCGGCACCTGGCAT-3′); β-actin specific primer pair, forward (5′-TGACAGGA TGCAGAAGGAGA-3′) and reverse (5′-GCTGGAAGGTGGACAGTGAG-3′) primer set (*Saito et al., 2019*). PCR was carried out using TaKaRa *Ex Taq* Hot Start Version (Takara bio).

## Gamete fusion assay

Superovulation was induced in each B6D2F1 female mouse (>8 weeks old) by injecting 7.5 IU of equine chorionic gonadotropin (eCG) and 7.5 IU of human chorionic gonadotropin (hCG) at 48 hr intervals. Oocytes were collected from the oviduct 16 hr after hCG injection, and placed in 50 µl of Toyoda, Yokoyama and Hoshi (TYH) medium. The zona pellucida was removed from the oocytes via treatment with 1.0 mg ml$^{-1}$ of collagenase (FUJIFILM Wako Pure Chemical Corporation) (*Yamatoya et al., 2011*). They were preloaded with 1 µg ml$^{-1}$ Hoechst 33342 (Thermo Fisher Scientific) in TYH medium for 10 min, and were washed prior to the addition of the spermatozoon. *Dcst1/2$^{-/-}$* spermatozoa were collected from the cauda epididymis, and capacitated *in vitro* for 2 hr in a 200 µl drop of TYH medium. Zona-free oocytes were co-incubated with $2 \times 10^5$ mouse spermatozoa ml$^{-1}$ for 30 min at 37°C in 5% $CO_2$. After 30 min of incubation, the eggs were observed under a fluorescence microscope after being fixed with 0.25% glutaraldehyde in flushing holding medium (FHM) medium. This procedure enabled staining of fused sperm nuclei only, by transferring the dye into spermatozoa after membrane fusion.

## Fluorescence imaging

Zona-free oocytes were prepared as described above. They were incubated with 0.5 µg ml$^{-1}$ MZ–FITC and 0.25 µg ml$^{-1}$ TH6–Alexa647 for 1 hr at 37°C in TYH medium. *Dcst1/2$^{-/-}$* spermatozoa were collected from the cauda epididymis, and capacitated *in vitro* for 2 hr in a 200 µl drop of TYH medium with 0.5 µg ml$^{-1}$ Mab125–Alexa546 that was covered with mineral oil (Merck). Then, $2 \times 10^5$ spermatozoa ml$^{-1}$ was inseminated in the oocytes for 30 min at 37°C with 5% $CO_2$. After the eggs were washed three times in TYH medium by transferring spots of oocytes containing media, the eggs were fixed with 4% paraformaldehyde and 0.5% polyvinylpyrrolidone (PVP) in phosphate buffered saline (PBS) for 30 min at room temperature, then observed in FHM medium on the glass-bottom dishes (No. 0, MatTek). A 100× oil-immersion objective (numeric aperture 1.49) was used to capture confocal images with an A1R microscope (Nikon). The pinhole was set at 3.0 airy unit. For 3D reconstruction, 100–120 fluorescent images were taken at 1 µm intervals on the Z axis, and then 3D images were reconstructed using the built-in software NIS-Elements ver. 4.3 (Nikon).

## *In vitro* fertilization

Mouse spermatozoa were collected from the cauda epididymis, and were capacitated *in vitro* for 2 hr in a 200 µl drop of TYH medium that was covered with mineral oil. Superovulation was induced in female mouse, as described above. Oocytes were collected from the oviduct placed in 100 µl of TYH medium, and incubated with $2 \times 10^5$ spermatozoa ml$^{-1}$ at 37°C with 5% $CO_2$. After 5 hr of insemination, the oocytes were washed with TYH medium and the fertilization rate was evaluated at the two-cell stage embryo.

## Production of SPACA6 polyclonal antibody

DNA coding for the extracellular region of mouse SPACA6$_{46-139}$ was constructed using synthetic DNA with codon use that was optimized for expression in *Escherichia coli*. The SPACA6$_{46-139}$ fragment was amplified by PCR with a 3'-primer with sequence encoding Ala-Gly-Gly-His-His-His-His-His-His, a linker plus a hexahistidine tag. The amplified PCR products were cloned back into pAED4 (*Doering and Matsudaira, 1996*). The construct was expressed using *E. coli* strain BL21 (DE3) pLysS (Agilent Technologies). The expressed SPACA6$_{46-139}$ fragment was accumulated in inclusion body. Inclusion body from 0.8 l culture was solubilized by the addition of 5 g of solid guanidine hydrochloride (GdnHCl), followed by the addition of 1 ml of 1M Tris-HCl (pH 8.5). The sample was subjected to HiTrap metal-chelating column (cytiva), which was equilibrated with 6 M urea, 10 mM Tris-HCl (pH 8.5), and 0.5 M NaCl, and eluted using imidazole gradient. Then, the sample was dialyzed against 10 mM Tris-HCl (pH 8.5) for a few days, filtered using a 0.45 μm filter (Merck Millipore), and concentrated by Amicon Ultra 10 (Merck Millipore). Protein stock concentration was determined by measuring the optical density at 280 nm based on the absorption of Trp and Tyr residues using the Edelhoch spectral parameters (*Gill and von Hippel, 1989*).

The SPACA6$_{46-139}$ fragment was mixed and emulsified with Freund's complete or incomplete adjuvant (BD Difco) and used for immunization of rabbits. Immunization was performed four times at 7-day intervals, and booster immunizations were administered before drawing blood. After purifying the anti-serum, antibody was kept at −80℃ until use.

## Western blot analysis

Mouse spermatozoa were collected from the cauda epididymis and were solubilized with 1% Triton X-100 and PBS with a protease inhibitor cocktail (nacalai tesque). Samples were treated with SDS-sample buffer (62.5 mM Tris-HCl [pH 6.8], 2% SDS, 50 mM dithiothreitol, 10% glycerol), then used for western blotting under reduced conditions. The membranes were probed with primary antibodies (1 μg ml$^{-1}$ α-IZUMO1 [Mab18], 1:500 α-SPACA6, and 0.2 μg ml$^{-1}$ α-BASIGIN [sc-9757]) followed by secondary antibodies conjugated to horseradish peroxidase (HRP) (Jackson ImmunoResearch Laboratories). Chemiluminescence reactions were performed with ECL Prime (cytiva).

## Statistical analyses

All statistical analyses were performed using OriginPro 2020b (Light Stone).

# Acknowledgements

We thank Dr. Kazuo Yamagata at Kindai University and the Biotechnology Research and Development (non-profit organization) at Osaka University for their technical assistance in ICSI and making *Dcst1/2* knockout mice, respectively.

# Additional information

## Funding

| Funder | Grant reference number | Author |
|---|---|---|
| Japan Society for the Promotion of Science | JP18H02453 | Naokazu Inoue |
| Japan Society for the Promotion of Science | JP17K07311 | Ikuo Wada |
| Takeda Science Foundation | | Naokazu Inoue |
| Gunma University | 19014 | Naokazu Inoue |

The funders had no role in study design, data collection and interpretation, or the decision to submit the work for publication.

## Author contributions
Naokazu Inoue, Conceptualization, Resources, Data curation, Formal analysis, Supervision, Funding acquisition, Validation, Investigation, Visualization, Methodology, Writing - original draft, Project administration, Writing - review and editing; Yoshihisa Hagihara, Resources, Investigation, Methodology, Writing - review and editing; Ikuo Wada, Conceptualization, Supervision, Funding acquisition, Writing - review and editing

## Author ORCIDs
Naokazu Inoue https://orcid.org/0000-0002-6187-8293
Yoshihisa Hagihara http://orcid.org/0000-0002-5980-1764
Ikuo Wada http://orcid.org/0000-0001-5668-6994

## Ethics
Animal experimentation: All animal studies were approved by the Animal Care and Use Committee of Fukushima Medical University, Japan (approval number: 2020017), and performed under the guidelines and regulations of Fukushima Medical University.

## Decision letter and Author response
Decision letter https://doi.org/10.7554/eLife.66313.sa1
Author response https://doi.org/10.7554/eLife.66313.sa2

# Additional files

## Supplementary files
• Transparent reporting form

## Data availability
All data generated or analysed during this study are included in the manuscript.

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
