## [Decision Letter]

**Acceptance summary:**

This is important scientific work that shows animal fertilization to be a process where there is conservation of certain proteins involved in sperm-egg events that occur just before fusion. This conservation is observed despite the dramatic differences in the cytology of fertilization observed in nematodes, insects and mammals.

**Decision letter after peer review:**

Thank you for submitting your article "Evolutionary-conserved sperm factors, DCST1 and DCST2, are required for gamete fusion" for consideration by *eLife*. Your article has been reviewed by 2 peer reviewers, one of whom is a member of our Board of Reviewing Editors, and the evaluation has been overseen by Anna Akhmanova as the Senior Editor. The following individual involved in review of your submission has agreed to reveal their identity: Andrew Singson (Reviewer #2).

Summary:

The current work under consideration for *eLife* conclusively shows that knockout of the two DC-STAMP protein-encoding genes studied in *C. elegans* and *Drosophila* cause completely penetrant sterility in mice. This is a major advance and is currently the best example showing the surprising conservation of the fertilization machinery across nearly 1 billion years of animal evolution.

Essential revisions:

Please complete the following revisions to make your manuscript acceptable for publication in *eLife*:

Figure 1 of the paper shows protein alignments of DC-STAMP proteins DCST1 and DCST2 with humans, mouse, worms (SPE-42 and SPE-49) and the fly (Snky) (Figure 1A). It also suggests that DCTS1 is more related to SPE-49 and Snky while DCST2 is more related to SPE-42 and fly "DCST2" (Figure 1B). There is no description of how these relationships are determined in the text or methods. This should be added to the results and Discussion section. A maximum likelihood tree analysis and diagram could be useful to understanding the evolutionary relationship between these genes. Was there a gene duplication and then DCST1 and DCST2 develop non-redundant functions?

Figure 1C Shows the tissue and age-dependent expression of Dcst1 and Dcst2 genes. Does the late testes expression correspond to the formation of mature sperm?

A better explanation of the TG constructs is needed in the Figure 2 and 3 legends. Are these cDNA constructs and, if so, could the lack of introns explain some of the issues related to their expression? This should also include why the authors used a heterologous promoter.

Western blot analysis showed that loss of IZUMO1 leads to a loss of SPACA6. Loss of DTSC1/2 as well as single gene loss also leads to loss of SPACA6 suggesting these genes function in a common fertility pathway. Loss of SPACA6 and DTSC1/2 does not alter IZUMO1 expression. Is it known where SPACA6 accumulates in sperm and during wild-type sperm-egg interactions? If so, please tell the reader.

Figure 3D, what is being depicted in this histogram? Is it evidence that sperm have penetrated the egg or that a viable zygote or two celled embryo was observed. In the case of *Drosophila*, sneaky mutants allow sperm to sort of penetrate the egg, but syngamy does not follow. Do the numbers in the mutants represent two-celled embryos? If so, state so explicitly in the Figure Legend and do not make the reader go to the Materials and methods. The results depicted in Figure 3F are quite convincing that Dcst1 and Dcst2 are nonredundant *in vivo*, which a far more powerful statement than the IVF results.

In looking at video one, it appears that KO of Dcst1 results in eggs that have produced the second polar body. However, the Dcst2 KO eggs have not produced a second polar body by this 6 hour post insemination time point. Is this the case? Please explain and note that if one allows polar body formation and the other does not allow it, this should be discussed.

Reviewer #1:

"Evolutionary-conserved sperm factors, DCST1 and DCST2, 1 are required for

gamete fusion" by Naokazu Inoue, Yoshihisa Hagihara and Ikuo Wada is a straightforward and relatively brief, but data-packed, description of the role of two DC-STAMP domain containing proteins in mouse fertilization. The emerging picture is that the proteins required for the steps that culminate in sperm-egg fusion show surprising conservation. The first clear case was reported by Nishimura et al. (2015) Curr. Biol. 25: 3225 where they showed a portion of mouse Izumo1 protein can functional replace its SPE-45 ortholog in C.elegans and the resulting chimeric protein allows fertility in nematodes. A shortcoming of that work was that no clear connection of Izumo1/SPE-45 was established in insects. Earlier work in both *C. elegans* and *Drosophila* established that DC-STAMP proteins are important in their respective organisms and showed that promising homologs expressed in testes existed in mammals. The current work conclusively shows that knockout of the two DC-STAMP protein-encoding genes studied in *C. elegans* and *Drosophila* cause completely penetrant sterility in mice. This is a major advance and is currently the best example showing the surprising conservation of the fertilization machinery across nearly 1 billion years of animal evolution.

Reviewer #2:

Fertilization requires several steps of sperm-egg interactions that culminate in fusion. Previously, mouse OC/DC-STAMP proteins were shown to be required for osteolastogenesis when mononucleated osteoclasts fuse to form multinucleated osteoclasts. In parallel, the *C. elegans* spe-42 and spe-49 genes were shown to encode multi-pass transmembrane proteins containing DC-STAMP "domains". Loss of function in either spe-42 or spe49 leads to male infertility. These mutants belong to what is termed the spe-9 class of genes where mutant sperm have normal morphology, motility, migration to the site of fertilization and contact with oocytes, yet fusion/fertilization of gametes does not occur. In this manuscript Inoue et al. identify DC-STAMP domain-containing homologues of spe-42 and spe-49 as well as the *Drosophila* Sneaky (Snky) gene. These proteins are called DCST1 and DCST2. The authors created knockout mice for DCST1 and DCST2 and DCST_1/2_ double knockout mice. In all cases males are completely or almost completely sterile with essentially the same mutant phenotype as the worm mutants (morphologically normal sperm that cannot fuse with the plasmamembrane of the egg. The authors also explore potential interactions with previously known mammalian fertilization molecules and find that DCST_1/2_ status impacts the stability of the sperm immunoglobulin superfamily protein SPACA6. This paper will be of highly significant impact since it shows conserved sperm molecular functions with clear genetic validation between invertebrates (fly and worm) and mammals separated by about 700 million to one billion years of evolution from a common ancestor.

---

## [Author Response]

Essential revisions:Please complete the following revisions to make your manuscript acceptable for publication in eLife:Figure 1 of the paper shows protein alignments of DC-STAMP proteins DCST1 and DCST2 with humans, mouse, worms (SPE-42 and SPE-49) and the fly (Snky) (Figure 1A). It also suggests that DCTS1 is more related to SPE-49 and Snky while DCST2 is more related to SPE-42 and fly "DCST2" (Figure 1B). There is no description of how these relationships are determined in the text or methods. This should be added to the results and Discussion section. A maximum likelihood tree analysis and diagram could be useful to understanding the evolutionary relationship between these genes. Was there a gene duplication and then DCST1 and DCST2 develop non-redundant functions?

Thank you for your comment. We have performed a maximum likelihood tree analysis, according to your suggestion, and have included it as our new Figure 1C. The explanations regarding this are described on P3, L1–4.

Unfortunately, we do not have the data to explain how DCST1 and DCST2 developed non-redundant function; however, this is an interesting point that we will examine further in future studies.

Figure 1C Shows the tissue and age-dependent expression of Dcst1 and Dcst2 genes. Does the late testes expression correspond to the formation of mature sperm?A better explanation of the TG constructs is needed in the Figure 2 and 3 legends. Are these cDNA constructs and, if so, could the lack of introns explain some of the issues related to their expression? This should also include why the authors used a heterologous promoter

Yes, “haploid-specific expression” in the original text (P2, L22) implies the late testes expression.

Accordingly, we have added explanations of TG constructs to the Figure 2 legend (P8, L8–12). The reason for choosing a heterologous *Calmegin* promoter was that we had been successful in testis-specific (pachytene spermatocyte to spermatid stage) expression of IZUMO1-TG (Inoue et al., 2005; PMID: 15759005).

Western blot analysis showed that loss of IZUMO1 leads to a loss of SPACA6. Loss of DTSC1/2 as well as single gene loss also leads to loss of SPACA6 suggesting these genes function in a common fertility pathway. Loss of SPACA6 and DTSC1/2 does not alter IZUMO1 expression. Is it known where SPACA6 accumulates in sperm and during wild-type sperm-egg interactions? If so, please tell the reader.

Thank you for your thoughtful comments. Unfortunately, there is no further description in the literature. Indeed, we had tried cell-oocyte binding assay using COS-7 cells expressing SPACA6-cfSGFP2 instead of spermatozoa, but there was no reaction (Inoue et al., 2015; PMID: 26568141).

Figure 3D, what is being depicted in this histogram? Is it evidence that sperm have penetrated the egg or that a viable zygote or two celled embryo was observed. In the case of Drosophila, sneaky mutants allow sperm to sort of penetrate the egg, but syngamy does not follow. Do the numbers in the mutants represent two-celled embryos? If so, state so explicitly in the Figure Legend and do not make the reader go to the Materials and methods. The results depicted in Figure 3F are quite convincing that Dcst1 and Dcst2 are nonredundant *in vivo*, which a far more powerful statement than the IVF results.

We appreciate this comment. We have added an explanation to the Figure 3D legend, accordingly (P10, L20–21).

In looking at video one, it appears that KO of Dcst1 results in eggs that have produced the second polar body. However, the Dcst2 KO eggs have not produced a second polar body by this 6 hour post insemination time point. Is this the case? Please explain and note that if one allows polar body formation and the other does not allow it, this should be discussed.

We apologize for the confusion. The first half of the Video 1 shows a heterozygous mutant of DCST1 and DCST2 double knockout spermatozoa, whereas the second half is a homozygous mutant of that. To avoid confusion, we have revised the Video 1 subheading.